# Complete Whole Genome Sequences of *Escherichia coli* Surrogate Strains and Comparison of Sequence Methods with Application to the Food Industry

**DOI:** 10.3390/microorganisms9030608

**Published:** 2021-03-16

**Authors:** Dustin A. Therrien, Kranti Konganti, Jason J. Gill, Brian W. Davis, Andrew E. Hillhouse, Jordyn Michalik, H. Russell Cross, Gary C. Smith, Thomas M. Taylor, Penny K. Riggs

**Affiliations:** 1Department of Animal Science, Texas A&M University, College Station, TX 77843-2471, USA; therrienda@gmail.com (D.A.T.); jason.gill@tamu.edu (J.J.G.); jordyn.michalik@tamu.edu (J.M.); hrcross@tamu.edu (H.R.C.); gary.smith@tamu.edu (G.C.S.); 2Texas A&M Institute for Genome Sciences and Society, MS 2470, College Station, TX 77843-2470, USA; k.kranti.neu@gmail.com (K.K.); hillhouse@tamu.edu (A.E.H.); 3Department of Veterinary Integrated Biosciences, Texas A&M University, College Station, TX 77843-4461, USA; bdavis@cvm.tamu.edu

**Keywords:** bacterial surrogate, *Escherichia coli*, whole genome sequence, short reads, long reads, closed genome, high throughput sequencing

## Abstract

In 2013, the U.S. Department of Agriculture Food Safety and Inspection Service (USDA-FSIS) began transitioning to whole genome sequencing (WGS) for foodborne disease outbreak- and recall-associated isolate identification of select bacterial species. While WGS offers greater precision, certain hurdles must be overcome before widespread application within the food industry is plausible. Challenges include diversity of sequencing platform outputs and lack of standardized bioinformatics workflows for data analyses. We sequenced DNA from USDA-FSIS approved, non-pathogenic *E. coli* surrogates and a derivative group of rifampicin-resistant mutants (rif^R^) via both Oxford Nanopore MinION and Illumina MiSeq platforms to generate and annotate complete genomes. Genome sequences from each clone were assembled separately so long-read, short-read, and combined sequence assemblies could be directly compared. The combined sequence data approach provides more accurate completed genomes. The genomes from these isolates were verified to lack functional key *E. coli* elements commonly associated with pathogenesis. Genetic alterations known to confer rif^R^ were also identified. As the food industry adopts WGS within its food safety programs, these data provide completed genomes for commonly used surrogate strains, with a direct comparison of sequence platforms and assembly strategies relevant to research/testing workflows applicable for both processors and regulators.

## 1. Introduction

Over recent decades, the landscape of food safety has undergone paradigm shifts as technological advancements in genomics enabled implementation of numerous measures for ensuring a safe and secure food supply [1]. Various methodologies (e.g., pulsed-field gel electrophoresis (PFGE), serotyping, phage typing, multi-locus sequence typing (MLST), etc.) have been utilized for identification and characterization of foodborne pathogens at the clinical level. While these techniques played invaluable roles in food safety, technological limitations prevent the resolution necessary for differential identification of closely related bacterial strains. Moreover, illnesses attributed to foodborne pathogens continue to persist and are estimated to be responsible for approximately 48 million illnesses (1 in 6 people), 128,000 hospitalizations, and 3000 deaths each year within the United States [2,3,4,5,6]. However, rapid technological advancements and drastic reductions in cost have made applications such as whole genome sequencing (WGS) via high throughput sequencing (next generation sequencing (NGS) and 3rd generation sequencing) appealing alternatives [7,8,9].

Currently, WGS is achieved via two types of sequencing methods that can be distinguished by the length of sequence fragments or “read lengths” (i.e., short- and long-read) produced [8,10]. Short-read sequencing platforms, such as those manufactured by Illumina, utilize massively parallel sequencing that yields read lengths of about 100 to 300 base pairs (bps) with a high level of accuracy. Typically, error rates in nucleotide identification (base calling) are less than 1% and result in 95% coverage of most bacterial genomes [8,10,11,12,13]. The short-read approach allows a researcher to make comprehensive estimations regarding the total number of genes present within the organism of interest, their classification in relation to other species, and the overall relatedness of their distinct gene sets to other organisms. While highly informative and effective for comparative gene-based studies, this technique is inadequate for producing sequences that span long repetitive genomic regions and large areas that are prone to rearrangement (e.g., deletions, insertions, repeats, and inversions). This limitation frequently results in incomplete genomic assemblies (draft genomes) of contiguous segments (contigs) that are oriented incorrectly or contain other structural errors due to repetitive elements and other genome features [8,10,12,13,14]. In contrast, long-read sequencing platforms can generate read lengths ranging from ~1000 bps to hundreds of kilobases in a single read. Unfortunately, the increased sequence length is offset by a significant reduction in sequence accuracy, with base calling error rates ranging from ~5–40% [8,11,13,14,15,16,17].

Despite the differences among WGS technologies, sequencing-based approaches consistently provide greater resolution and discriminatory power for distinguishing closely related bacterial species compared to previous methods, thus improving foodborne pathogen surveillance systems and trace back investigations [4,5,18,19,20,21,22]. WGS datasets can be simultaneously used in multiple investigative analyses (e.g., subtyping, antibiotic resistance profiling, virulence genetic markers, screening of mobile genetic markers, etc.) or stored for future analyses. For these reasons, WGS is being adopted by federal regulatory and public health related entities (e.g., Center for Disease Control (CDC), Food and Drug Administration (FDA), USDA-FSIS) as one of the primary methods for surveillance, outbreak investigations, and the tracing of transmission routes of foodborne pathogens [8,18,22,23].

The technological benefits of WGS support adoption by government and regulatory agencies, but certain aspects must be addressed before WGS can be widely incorporated as routine screening within the food industry, if at all. Arguably, one area of greatest difficulty pertaining to this technology is the abundant diversity in workflows that exist for processing and sequencing of the samples, as well as bioinformatic analyses and interpretation of large volumes of data. Genomic technologies have undergone rapid advancements that enabled innovations and accessibility but have also resulted in a large variety of preparatory workflow procedures, sequencing platforms with diverse utility, and innumerous bioinformatic analytical tools [8,10,13,18,22,24,25].

The overall objective of this project was to produce high quality, complete genome assemblies for a group of USDA-FSIS-approved non-pathogenic *E. coli* surrogates (ATCC BAA-1427, BAA-1428, BAA-1429, BAA-1430, and BAA-1431) and annotate any virulence factor genes or subunit genes present in the genomes. To also provide direct comparison of the technologies, we conducted comparative analyses of the short-read and long-read sequences for direct comparison to the hybrid approach. We also annotated the genes conferring rifampicin resistance in derivative isolates at the same time. The USDA-FSIS has previously supported the use of non-pathogenic surrogate organisms for the validation of in-plant intervention strategies to reduce the presence of foodborne pathogens. This model of intervention is highly beneficial because it allows one to determine the efficacy of a current intervention strategy without inherent risk of contaminating testing equipment or facilities. This group of surrogates is of particular interest because they been shown to possess similar properties to pathogenic *E. coli* and *Salmonella enterica* and are widely used in contemporary research and intervention validation [26,27,28,29,30,31]. Despite their widespread use, genetic information for these strains is not readily available. Thus, the data presented here contribute to the existing body of knowledge regarding sequencing approaches for detection of genes associated with pathogenesis and antibiotic resistance. In addition, these data provide completed genomes for these widely used surrogates that will be an invaluable resource for processors and regulatory officers in differentiating these strains from pathogenic strains of *E. coli* and supporting decision-making for the incorporation and application of WGS for food safety applications.

## 2. Materials and Methods

### 2.1. Bacterial Surrogates

Five non-pathogenic *E. coli* biotype I strains (isolates BAA-1427, BAA-1428, BAA-1429, BAA-1430, and BAA-1431) were obtained from the American Type Culture Collection (ATCC; Manassas, VA, USA) and revived according to ATCC guidance [32]. The surrogates originated as isolates from cattle hides at facilities in the Department of Animal Science at Iowa State University (Ames, IA, USA) [29,33]. The strains were confirmed to lack antibiotic resistance or a subset of known virulence factors by the *E. coli* Reference Center of Penn State University (University Park, PA, USA) and underwent further toxin testing via the application of commercial kits as well as tissue culture testing (African green monkey kidney (Vero) cells) by the depositor [33]. Isolate details are described in tables available at the ATCC website [32,34]. These *E. coli* isolates were propagated twice in 5.0 mL of tryptic soy broth (TSB; Becton, Dickinson and Co., Sparks, MD, USA) (24 h, 35 °C), and then grown on tryptic soy agar (TSA; Becton, Dickinson and Co., Sparks, MD, USA) slants, TSA Petri plates, TSA + rifampicin (100.0 mg/L; TSA-R) plates, and MacConkey agar (MAC; Becton, Dickinson and Co., Sparks, MD, USA) Petri plates (24 h, 35 °C). Following overnight incubation, colonies of parent *E. coli* isolates grown on the TSA slants and streaked on plates were verified as rifampicin-sensitive or rif^R^. API^®^ 20E (bioMérieux, Inc. N.A., Durham, NC, USA) tests were used to identify organisms as *E. coli* according to manufacturer guidance. Additionally, including in the sequencing experiment were three rif^R^ mutants, coded BAA-1427 rif^R^, BAA-1428 rif^R^, and BAA-1430 rif^R^ [35,36]. Following verification of parent strain identities, these mutants were prepared from isolates BAA-1427, BAA-1428, and BAA-1430 by the T.M. Taylor lab (Texas A&M University, College Station, TX, USA) and made available for this project. The rif^R^ strains were verified as described above with respect to *E. coli* identification, and via overnight growth on TSA-R media (24 h, 35 °C).

### 2.2. DNA Extraction and Quantification

For DNA extraction, *E. coli* (parents, rif^R^ mutants) were propagated from working stocks incubated in 5.0 mL TSB (24 h, 35 °C) as previously described. TSA streak plates were created from each bacterial isolate and incubated likewise (24 h, 35 °C). A single bacterial colony was selected and grown in 5.0 mL TSB (24 h, 35 °C) for DNA extraction. The samples were centrifuged at 10,000 g for 2 min and cell pellets were frozen at −80 °C until used. A phenol/chloroform DNA extraction protocol was used to isolate genomic DNA from cell pellets [37]. The extraction procedure was modified with the substitution of 1-bromo-3-chloropropane (BCP; Molecular Research Center Inc., Cincinnati, OH, USA) for 24:1 chloroform:isoamyl alcohol prior to ethanol precipitation [38]. DNA samples were quantified via spectrophotometry (NanoDrop ND-1000, Thermo Fisher Scientific, Waltham, MA, USA), visualized by electrophoresis through a 1.0% agarose SFR gel (AMRESCO, Solon, OH, USA) in a 1× Tris/Borate/EDTA (TBE) buffer solution, and stained with SYBR green (Sigma-Aldrich Corp., St. Louis, MO, USA).

### 2.3. Genomic Sequencing

Bacterial genomic DNA was sequenced at the Texas A&M Institute for Genome Sciences and Society (TIGSS) core facility (College Station, TX, USA) by Illumina MiSeq and Oxford Nanopore MinION gene sequencing platforms (Illumina, Inc., San Diego, CA, USA and Oxford Nanopore Technologies, Oxford, UK). The derivative group of rif^R^ mutants were sequenced only via MiSeq. Prior to sequencing, the bacterial DNA was re-quantified via the Qubit 2.0 Fluorometer (Thermo Fisher Scientific, Waltham, MA, USA) as recommended by the Illumina MiSeq and Oxford Nanopore MinION library preparation kit protocols. Libraries were prepared with the Nextra XT v2 library preparation kit (Illumina, Inc., San Diego, CA, USA) for the Illumina MiSeq platform, and the Rapid Barcoding Kit (SQK-RBK004, Oxford Nanopore Technologies, New York, NY, USA) for the MinION. Quality of all sample libraries was evaluated via the Agilent 2200 TapeStation (Agilent, Santa Clara, CA, USA) prior to sequencing

### 2.4. Genome Assembly

Upon completion of sequencing reactions, the raw sequence data were downloaded from the Illumina BaseSpace (Illumina, Inc., San Diego, CA, USA) and Oxford Nanopore Metrichor (Oxford Nanopore Technologies, Oxford, UK) cloud-based storage systems and uploaded onto the TIGSS High Performance Computing Cluster for further processing. The sequence data produced by the MinION were converted from the FAST5 to FASTQ format via the Oxford Nanopore Albacore v2.0.1 base caller [39]. Sequence quality was assessed via FastQC, and low-quality sequence data and the adapter sequences were removed with Trimmomatic v0.32 [40,41]. The SPAdes software tool (v3.13.0) was used to generate a short-read assembly from the MiSeq data, and the Canu v2.0 single-molecule sequence assembler was used to generate long-read assembly from the MinION data [42,43]. Once assembled the contigs for each bacterial sample were screened and those contigs that were <1000 bps (MinION), <500 bps (MiSeq), possessed low coverage scores, and/or were poorly associated with *E. coli* species were removed from the assemblies. After low-quality contigs had been removed, sequence statistics were calculated for each sample, with overall rates of coverage being calculated via the BEDTools software [44].

### 2.5. Polishing and Error Correction

Raw unfiltered MiSeq reads and the Canu FASTA long-read assemblies were combined into hybrid assemblies using the Unicycler genomic assembler [45]. During this process, the generated hybrid assemblies underwent various cycles of polishing and error correction using the integrated Pilon software tool v1.23 [46]. Following this the degree of completeness of each hybrid genome was assessed using the Benchmarking Universal Single-Copy Orthologs (BUSCO) software v4 and was compared with the lineage enterobacteriales (composed of 216 species and 781 orthologs) [47]. To further close these genomes, each was processed using the reference-guided contig ordering and orienting tool (RaGOO) with the *E. coli* K12 substr. MG1655 (NC_000913.3) reference genome [48]. Lastly, for the samples that were not reduced to a single contig, analysis was conducted via BLAST [49] to align the nucleotide sequences of the surplus contigs to known sequence to identify their origins (BLASTn).

### 2.6. Virulence Factor Screening and Verification

Serotyping and MLST for the three assemblies of each bacterial surrogate was determined with the open access SeroTypeFinder 2.0 and MLST 2.0 software [50,51]. The generated assemblies for each of the *E. coli* surrogates were analyzed by translated BLAST analysis (BLASTx) against a dataset of *E. coli* virulence factors extracted from the Virulence Factor Database (VFDB) using an e-value cutoff of 10^−5^ (Appendix A) [52,53,54,55]. The genome of the known non-pathogenic *E. coli* str. Nissle 1917 (NZ CP022686) was used as a control to filter spurious hits by subtraction of BLASTx hits shared between the surrogates and Nissle with the remaining factors undergoing further investigation. Analyses were conducted on the Texas A&M University Center of Phage Technology (CPT, College Station, TX, USA) Galaxy instance [56].

The remaining detected virulence factors were examined to confirm the presence of complete genes and/or gene modules as appropriate. Individual bacterial contigs were opened with Sanger Artemis (v. 18.0.0, Wellcome Sanger Institute, Cambridge, UK) and the regions containing suspected virulence determinants based on BLASTx coordinates were manually annotated and their protein sequences compared to those of known functional virulence factors by BLASTp to determine if they were complete and free of alterations that may render them non-functional [57]. The percent identities for each of the potential pathogenic elements found within the hybrid assemblies were calculated using the Sørensen-Dice coefficient [58,59].
(1)SDC=2|x∩ y||x|+|y|

The rif^R^-mutants were excluded from this analysis, as it was expected that their matches would correspond with their surrogate parent strains.

### 2.7. Detection of known rpoB Rifampicin Resistance Mutations

The *rpoB* DNA sequences of the parental surrogates BAA-1427, BAA-1428, and BAA-1430 assemblies (i.e., long-read, short-read, and hybrid), and their corresponding short-read rif^R^-mutants BAA-1427 rif^R^, BAA-1428 rif^R^, and BAA-1430 rif^R^ counterparts were compared to that of *E. coli* str. K-12 substr. MG1655 (NC_000913.3) via BLASTn to detect mutations commonly associated with rifampicin resistance [60].

## 3. Results

### 3.1. Comparison of Sequence Assembly Statistics

Assembly summaries for the MinION and MiSeq assemblies were calculated and compared along with their Serotypes and MLSTs (Table 1 and Table 2). Sequence generated from the Oxford Nanopore MinION platform resulted in read lengths that were approximately 10-fold longer than read outputs from the Illumina MiSeq sequencer (Table 1 and Table 2). The longer read lengths enabled assembly of sequence reads into fewer and longer contiguous stretches (contigs) and resulted in greater overall genome coverage for each bacterial sample (Table 1). The draft assemblies produced from MinION data resulted in a large singular contig for each assembly (4–5 Mbs) with a subset of smaller contigs averaging 1kb in size. In contrast, the MiSeq platform resulted exhibited greater uniformity in the size distribution of contigs, with the largest ranging from ~300–500 kbs with a steady decline in size to the smallest contig which was ~70 bps (Appendix A). These results are consistent with expected ranges for each platform, as a consequence of the unique chemistry and mechanisms for each technology. However, the total assembled lengths for each bacterial genome differed by only 100-300 kb between the MinION and MiSeq sequencing platforms, reflecting a 2–6% difference among surrogate counterparts (Table 1 and Table 2).

### 3.2. Hybrid Assembly Statistics and Analysis

The MinION and MiSeq assemblies were combined to improve the overall genome assembly of each surrogate. For each hybrid assembly, summary statistics were calculated and serotypes and MLSTs were identified for comparison with the long- and short-read counterparts. Total lengths of the hybrid assemblies for all five *E. coli* surrogates increased when compared with the MiSeq assemblies and slightly decreased when compared to that of the MinION assemblies (Table 1, Table 2 and Table 3). In considering the total number of contigs and the overall completeness of the genomes, significant improvements were observed in the hybrid assemblies in (Table 1, Table 2 and Table 3). In most cases the genomes were reduced to a single contig. The remaining additional contigs observed for two of the surrogates (BAA-1428 & BAA-1430) were identified via BLASTn to be residual fragments of existing plasmids. Additionally, when the hybrid genomes were compared with the lineage enterobacteriales (216 species and 781 orthologs) within BUSCO, each sample’s genome was reported to be between ~99.8 and 99.9% complete (Table 3). Lastly, the hybrid assembly’s quality was further improved compared with the other assemblies as it underwent multiple rounds of polishing via Pilon which resulted in numerous corrections within each genome (Table 3). GenBank Genomes database accession numbers [61] for each genomic assembly are included in Table 3, and each sample was annotated via the automated NCBI prokaryotic genome annotation pipeline [62].

For three of the bacterial genomes (BAA-1427, BAA-1429, and BAA-1431), each assembly was closed and reduced to a single observable contig that was within the range of a standard *E. coli* genome. The BAA-1428 genome contained one contig that was comparable in size with the three completed genomes, and a smaller 6762 bps. From BLASTn analysis, the smaller, non-chromosomal contig was found to be identical (100% coverage) to several plasmid sequences existing in public databases: *Salmonella enterica* serovar Newport plasmid pSNE1-1926 (CP025235.1) (6761 bps), *Salmonella enterica* serovar 1,4(5),12:i- plasmid p11-0813.1 (CP039594.1) (6760 bps), and *Salmonella enterica* serovar Enteritidis plasmid p4.4 (MG948564.1) (6760 bps). Of these, the proposed plasmid differed the most from *Salmonella enterica* serovar 1,4(5),12:i- plasmid p11-0813.1 (CP039594.1) by only 50 nucleotide alterations that existed primarily between nucleotides 1201–1315. Both *Salmonella enterica* serovar Newport plasmid pSNE1-1926 (CP025235.1) and *Salmonella enterica* serovar Enteritidis plasmid p4.4 (MG948564.1) possessed a nucleotide shift (A→G) at nucleotide 1371 when compared with the proposed BAA-1428 plasmid. Additionally, the proposed plasmid was compared with the other plasmids, they all possessed deletions within a region of low-complexity sequence (i.e., homopolymeric guanines) that spans between nucleotides 426–436. With the only notable differences existing within this region of low-complexity sequence and at nucleotide 1371 (A→G) it could not be determined if the proposed BAA-1428 plasmid was more similar to the *Salmonella enterica* serovar Newport plasmid pSNE1-1926 (CP025235.1) or *Salmonella enterica* serovar Enteritidis plasmid p4.4 (MG948564.1).

The result for BAA-1430 however was enigmatic when compared with the others as not only was it on average ~300 kbps larger than the other assembled genomes in total length but despite all further processing, remained at five observable contigs. Of these the largest was 4,988,672 bps in overall size which is more comparable to the other genomes. Four smaller contigs that were present ranged from 96,846, 9368, 6077, and 5649 bps in length. When BLASTn analysis was performed on these remaining non-chromosomal contigs it was found that the second contig (96,846 bps) displayed the highest genetic identity to the *E. coli fergusonii* plasmid pRHB23-C01_2 (CP057566.1; 99.74% identity, 83% coverage). The third contig (9368 bps) most resembled the *Serratia liquefaciens* plasmid pS12 (CP048786.1; 99.97% identity, 94% coverage. The fourth contig (6077 bps) shared a 100% identity and 100% coverage with the *E. coli* plasmid pRHB08-C23_3 (CP057955.1). Lastly, the fifth and smallest contig (5649 bps) revealed a 99.83% identity and 96% coverage when compared with an unnamed plasmid previously associated with *E. coli* strain RHB13-C21 (CP055721.1).

### 3.3. Virulence Factor Presence/Absence Determination and Characterization

The MinION, MiSeq, and hybrid genome assemblies from each of the five *E. coli* surrogates encoded genes associated with a subset of predicted regulatory protein adherence factors (Table 4). However, the genomes lacked many of the necessary genes that encode vital structure elements/subunits necessary for assembly of the full protein complexes, thus rendering these adherence factors non-functional (Table 4). The genomes assembled from MinION sequence were of lower resolution, containing multiple indels and higher errors rates that significantly reduced statistical confidence for detection of many of the adherence factor sequences that were examined, in comparison with those generated by the MiSeq and hybrid assemblies (Table 4). However, the MinION, MiSeq, and hybrid genome assemblies indicate that strains BAA-1427, and BAA-1431 encode complete cytolethal distending toxin (CDT) A, B, and C, and cytotoxic necrotizing factor 1 (CNF1) (Table 4). The percent identities of each of the identified pathogenesis factors for each hybrid assemblies were calculated with the Sørensen-Dice coefficient [58,59]. Predicted amino acid sequences identified as CDT A, -B, and -C within both surrogate sequences were 56.03%, 69.87%, and 40.56% similar to their functional CDT A, -B, and -C counterparts (GenBank: CAD48849.1, CAD48850.1, and CAD48851.1), respectively [63]. Additionally, the CNF1-like amino acid sequence in BAA-1427 and BAA-1430 possessed 53.48% percent identity to functional CNF1 (GenBank: CAA50007.1) [64].

### 3.4. Detection of Known rpoB Rifampicin Resistance Mutations

The *rpoB* DNA sequences of the parental surrogates BAA-1427, BAA-1428, and BAA-1430 assemblies (i.e., long-read, short-read, and hybrid), and their corresponding short-read rif^R^-mutants BAA-1427 rif^R^, BAA-1428 rif^R^, and BAA-1430 rif^R^ were compared to that of *E. coli* str. K-12 substr. MG1655 (NC_000913.3) to gauge each method’s utility for enabling detection of known mutations that confer rifampicin resistance (Appendix A). When screened, it was found that the three BAA-1427 assemblies (parent strains) and the BAA-1427 rif^R^ assembly (mutant child strain) shared a silent mutation (A206 to A), while the rif-resistant strain contained an additional L533 to P mutation. The BAA-1428 genomes and BAA-1428 rif^R^ shared a silent mutation (T486 to T), and BAA-1428 rif^R^ also possessed a mutation in S512 to P. Additionally, the parent BAA-1430 genomes and the BAA-1430 rif^R^ genome shared a series of silent mutations (P489 to P, L623 to L, and G846 to G) when compared to *E. coli* K-12. Lastly, in addition to those silent mutations the BAA-1430 rif^R^ mutant possessed an additional mutation of H526 to Y. However, it is of note that while the MinION assemblies did contain the same mutations as the MiSeq, hybrid, and rif-resistant assemblies, they also contained a large number of additional indels and were ultimately deemed unsuitable for the reliable identification of rif^R^–associated single-nucleotide polymorphisms (SNPs) (Appendix A). While elements known to confer rif^R^ could be detected in the MinION assemblies, this would not be a reliable approach for detecting novel mutations.

## 4. Discussion

We conducted WGS of a group of USDA-approved non-pathogenic *E. coli* surrogates via two popular NGS technologies and, also performed short-read sequencing on rif^R^ derivatives that exist for three of them. Our objective was to generate and characterize complete genome sequences for these important resources. At the same time, we used the opportunity to directly compare two common sequencing platforms and evaluate their usefulness for identification of potential pathogenic elements or known SNPs that confer rifampicin resistance. Both sequencing methods enabled production of draft genome assemblies for each bacterial strain, although key differences were apparent-notably in the distribution of contig size between the two platforms.

Despite producing draft genomes that typically contained less than 100 contigs of quality sufficient for comparative genomic analysis, with some exception to the MinION genomes due to high error rates, a complete, closed genome was not produced by either method alone (Table 1 and Table 2). However, sequence from the MinION enabled assemblies for each sample in which a single contig comprised ~94–99% of the total assembly length (Table 1). Consistent with previous findings, when the MinION and MiSeq assemblies were utilized in producing hybrid de novo assemblies, the unique strengths of each method combined to overcome their individual limitations [12,45,65,66,67]. The combined hybrid MinION and MiSeq assembly resulted in drastic quality improvements in each of the bacterial genomes assemblies (Table 3). The hybrid assembly was similar in overall length but had greatly reduced contigs and improved quality for bacterial assembly. Analysis of each hybrid for completeness (via BUSCO using lineage enterobacteriales), indicated that each genome assembly was ~99.8–99.9% complete (Table 3). Overall, the hybrid assemblies proved to be superior for closing the bacterial genomes and provide an invaluable tool for precisely distinguishing between multiple closely related species of interest.

For assessing pathogenesis, all three assembly strategies enabled identification of genetic sequence associated with various adherence factors and regulatory elements within all the isolates. Differences were observed between methods due to statistical cut-offs for identity established prior to the analysis (Table 4). On average the MinION genome assemblies lacked the same degree of resolution and confidence in predicting the presence of several of the adherence factors resulting in several false-negatives. The MinION assemblies also appeared to possess multiple frameshifts and duplications, further complicating virulence factor analysis. The hybrid assemblies resulted in more accurate representation of the genomes of these bacteria.

The genome assemblies for the surrogate strains were scanned for the presence of gene sequences that encode virulence factor subunits (Table 4; details of virulence factors provided in Appendix A). Although these lines were previously shown to lack functional virulence factors by other methods [34], the availability of these new complete genome assemblies enabled a more detailed investigation of the strains. Four strains (BAA-1427, BAA-1429, BAA-1430, and BAA-1431) possessed genetic sequences similar to those found within the enteropathogenic *E. coli* (EPEC) adherence factor plasmid (EAF) pB171. Sequences for bundle-forming pili (BFP) subunits BfpB (secretin), BfpE (inner membrane protein), and BfpH (transglycolase) [68] were identified in three MiSeq assemblies (BAA-1427, BAA-1430, and BAA-1431). Three of the hybrid and MinION counterparts (BAA 1427, BAA-1430, BAA 1431) lacked BfpH, and the MinION BAA-1430 lacked all three Bfp subunits. However, the noted absences following BLASTx analysis resulted from failure to meet the statistical threshold, likely reflecting nucleotide sequence variations. Additionally, all the assemblies for two strains (BAA-1429 and BAA-1430) encoded the BfpW/PerC transcriptional activators, which are part of the plasmid-encoded regulator (Per) responsible for BFP formation and activation of select genes within the locus of enterocyte effacement (LEE) [69,70,71,72]. Despite the presence of some subunits, these sequence elements are insufficient for formation of fully functional pili due to the absence of key accompanying subunit genes.

Apparent homologs of PapE (tip fimbriae), PapG (digalactoside-binding adhesion), and PapJ (assemble/integrity), were observed for all three assemblies of the BAA-1429 strain genome. These elements help comprise the pyelonephritis-associated pili/P fimbriae commonly seen in uropathogenic *E. coli* [73,74,75,76]. The pap operon is responsible for the formation of this pilus and has been previously described as encoding eleven distinct proteins (i.e., PapA, -B, -C, -D, -E, -F, -G, -H, -I, -J, -K). However, the presence of a fully functional P fimbriae pilus is unlikely due to the absence of fundamental structural and assembly elements [76,77,78,79].

Some form of a colonization factor (CF) that is typically observed in enterotoxigenic *E. coli* (ETEC) was observed in genome assemblies from all the strains. The most prevalent CF was the protein CsnA, which is a component of the major pilin monomer of the CS20 fimbriae [80,81,82]. However, in almost every instance, sequence similarity of these remained close to the statistical cut-off for identity, indicating lack of similarity to functional virulence factors. In addition to CFs, the BAA-1428 and BAA-1430 genomes contained genes similar to the CswA factor, commonly associated with the formation of the structural CS12 fimbriae subunits [81]. Lastly, *E. coli* BAA-1430 exclusively possessed sequence similarity to the CfaB, CooA, and CsbA CFs, associated with the colonization factor antigen I (CFA/I), CSI pilin major subunit, and the CS17 fimbrial subunit [81,83,84]. The CFs found within these genomes are associated with virulent ETEC; for pathogenesis to arise within these species there are two primary factors that must be present, which are the enterotoxins: heat-labile (LT) and/or heat-stable (ST) toxins, of which are frequently transported within the same plasmid as the CFs [81,84,85]. Their absence indicates these CFs may represent fimbriae/adherence factors that are not associated with pathogenesis, but confer similar structural properties to those that are. It is not uncommon that *E. coli* isolates, whether pathogenic or non-pathogenic, possess some mix of colonization and/or adherence factors [86,87,88]. However, in this experiment none of the identified sequences with resemblance to any putative virulence factor is expected to be functional or pose a hazard. The findings here indicate that the isolates of interest harbored no other detectable factors typically associated with virulent ETEC.

In two strain genomes (BAA-1427 and BAA-1430) potential homologous sequences encoding toxins associated with pathogenic *E. coli* were detected, but with only weak similarity -thus not likely functional. For example, the CNF1 holotoxin, an AB toxin, is documented to operate via RhoGTPases activity within eukaryotic cells but shared only 53.48% percent translated amino acid identity with the known functional toxin [89,90,91]. Documented regions within CNF1 that are responsible for host cell binding, as well the C-terminal portion that expresses the catalytic activity of this protein, were each ~50% identical in the surrogates when compared to the respective regions within their functional counterpart [92,93,94]. Additionally, the BAA-1427 and BAA-1430 genomes all appeared to harbor sequences sharing similarities with cytolethal distending toxin (CDT) [95,96]. CDT is a tripartite holotoxin responsible for cell cycle arrest and apoptosis within mammalian cells and is composed of a deoxyribonuclease-like toxin B-subunit and A and C subunits responsible for transporting the B subunit to the surface of the host cell [97,98,99,100,101]. Upon examination of these identified subunits, it was found that our samples only shared a 56.03% identity to subunit A (CAD48849.1), 69.87% to subunit B (CAD48859.1), and 40.56% to subunit C (CAD48851.1) when compared with functional toxins.

The availability of rifampicin-resistant strains derived from the surrogates also enabled screening for antibiotic resistant strains. All three sequencing and assembly methods enabled successful identification of SNPs within the *rpoB* gene known to confer rifampicin resistance [102,103]. As discussed, the genome assemblies generated from MinION data reflected the lowest quality assembly. While potentially useful for analysis of highly specific, known SNPS, these data would be difficult to utilize for discovery of new mutations (Appendix A). Two isolates (BAA-1427 and BAA-1427 rif^R^) appeared to share a silent mutation encoding amino acid A206, while the rif-mutant contained an additional L533 to P mutation, a change previously documented to confer rifampicin resistance [103]. Two strains (BAA-1428 and BAA-1428 rif^R^) shared a silent mutation (T486 to T) compared to *E. coli* K-12. BAA-1428 rif^R^ also possessed an additional S512 to P mutation, which has not been previously reported to confer rifampicin resistance but resides within the first cluster of the rifampicin resistant determining region (RRDR) [102]. Lastly, the BAA-1430 genomes and BAA-1430 rif^R^ both shared silent mutations (P489 to P, L623 to L, and G846 to G), and the BAA-1430 rif^R^ possessed a H526 to Y mutation, also documented to confer drug resistance [103]. Ultimately, each technique was useful for identification of key differences between the parent surrogate strains and the K-12 reference *rpoB* gene, but the long-read genomes lacked the precision to accurately distinguish consistent key differences that conferred rifampicin resistance due to a high concentration of nucleotide deletions and false amino acid shifts.

## 5. Conclusions

This study provided a direct comparison of two common sequencing platforms and discussion of genome assembly characteristics applied to surrogate bacterial strains for which genome sequences were not available. From both research and regulatory standpoints, the application of WGS and subsequent bioinformatic analyses are indeed the tools of the future - unifying many traditionally used microbiological analyses into a singular workflow and enabling greater precision in surveillance of foodborne pathogens for quicker and more efficient regulatory response to foodborne outbreaks. Institutes such as the Center for Genomic Pathogen Surveillance have already adopted and standardized WGS-based pathogen detection. Similar research and application in both academe and industry will continue to accelerate. Both long- and short-read sequencing methods serve valuable, yet distinct roles for construction of complete microbial genomes. Long-read capacity facilitates better genome assembly and reveals structural properties of the genome that are not readily sequenced by other means. Short-read methods improve precision and resolution required for investigative studies and certain targeted analyses. As demonstrated in this study, when combined in hybrid fashion, the two sequencing approaches together are invaluable in enabling completed high-quality genomes to be constructed and accessible within databases for utilization in traceback and recalls in the instance of foodborne outbreaks in which a high degree of resolution is required for distinguishing between closely related bacterial strains. Libraries of high quality, complete genomes also serve a valuable research function, and provide a resource for further understanding of genomic sequences or alterations that can confer pathogenesis.

The application of true WGS (i.e., production of a complete genome comprised of a single contiguous sequence) as a means for daily routine screening within a food processing facility’s food safety program is impractical and other forms of high-throughput sequencing may be more optimal. As demonstrated here, short-read platforms such as MiSeq provide a time- and cost-efficient means of simple of known pathogenic elements or antibiotic resistance genes. For the purposes of a food processing facility, confirmation of elements that confer pathogenesis is required, and while the methods described within this paper offered a means to achieve this, a fully closed bacterial genome is not required for most routine screening. While the cost of sequencing has greatly diminished as the quality of generated output continues to increase, WGS remains a data intensive process, relies on evaluation of DNA extracted from a pure bacterial culture, and is largely inefficient for routine screening. Ultimately, while routine WGS of samples taken within the food processing facility would serve a valuable means for differentiating what is being transported into the facility from the native microflora that pre-existed within the facility, it is impractical as a means for routine screening for outgoing lots. Although WGS enables sequence discrimination of genetic elements associated with both pathogenic and non-pathogenic strains, the current findings demonstrate that this outcome can be achieved more optimally via high-throughput targeted sequencing. Regardless, high-throughput sequencing methods will become increasingly important in food safety applications. These analyses enable insight into the genetic make-up of the surrogate strains studied that is useful for a variety of research applications and can also help inform decision-making for the incorporation and application of WGS within industry food safety programs.

## Figures and Tables

**Table 1 microorganisms-09-00608-t001:** Long-read Oxford Nanopore MinION assembly sequence statistics.

Bacterial Strains	O Type	H Type	MLST ^1^	Contigs	Assembled Length	Largest Contig	Average Coverage
BAA-1427	-	4	n/a	74	5,034,864 bps	4,743,343 bps	323.673×
BAA-1428	154	16	n/a	67	5,050,340 bps	4,806,641 bps	311.819×
BAA-1429	166	12	n/a	20	4,856,504 bps	4,816,131 bps	362.642×
BAA-1430	28ac/42	21	n/a	19	5,217,837 bps	5,022,067 bps	310.567×
BAA-1431	-	4	n/a	34	4,982,422 bps	4,753,397 bps	306.167×

^1^ MLST (Multi-locus Sequence Typing)–types could not be determined due to imperfect matches.

**Table 2 microorganisms-09-00608-t002:** Short-read Illumina MiSeq assembly sequence statistics.

Bacterial Strains	O Type	H Type	MLST	Contigs	Assembled Length	Largest Contig	Average Coverage
BAA-1427	-	4	10	91	4,825,300 bps	434,834 bps	51.211×
BAA-1428	154	16	165	127	4,758,825 bps	319,570 bps	57.141×
BAA-1429	166	12	10	87	4,739,915 bps	523,910 bps	60.601×
BAA-1430	28ac/42	21	278	103	5,009,161 bps	421,121 bps	49.422×
BAA-1431	-	4	10	91	4,829,685 bps	404,666 bps	47.608×

**Table 3 microorganisms-09-00608-t003:** Hybrid assembly sequence statistics.

Bacterial Strains	O Type	H Type	MLST	Pilon ^1^	BUSCO ^2^	Contigs	Assembled Length (bps)	Largest Contig (bps)	Average Coverage	GenBank Accession No.
BAA-1427	-	4	10	6	99.9%	1	4,886,306	4,886,306	152×	CP063979
BAA-1428	154	16	165	5	99.8%	2	4,876,786	4,870,024	151×	CP063956-CP063967
BAA-1429	166	12	10	4	99.9%	1	4,812,017	4,812,017	186×	CP063969
BAA-1430	28ac/42	21	278	8	99.9%	5	5,106,612	4,988,672	138×	CP063970-CP063974
BAA-1431	-	4	10	6	99.9%	1	4,889,455	4,889,455	135×	CP063958

^1^ Indicates the number of rounds of error correction each assembly underwent during Pilon processing.^2^ Indicates the predicted completeness of each assembly generated by BUSCO (Benchmarking Universal Single-Copy Orthologs) after comparison to the lineage enterobacteriales.

**Table 4 microorganisms-09-00608-t004:** Virulence attributes observed in bacterial surrogates. Subunits of virulence factors that were detected in each strain are indicated. An e-value limit of <0.00001 was adopted as a cut-off for protein identity (Blastx analysis). GenBank accession numbers for the virulence factors and their corresponding subunits within this table are provided in Appendix A.

Virulence Factors	BAA-1427	BAA-1428	BAA-1429	BAA-1430	BAA-1431
Bundle-forming pili subunits (BFP)	bfpB *^(-)^, bfpE *^(-)^, bfp^HI(-)^	-	-	bfpB ^IH(-)^, bfpE ^IH(-)^, bfpH ^IH(-)^	bfpB *^(-)^, bfpE *^(-)^, bfpH ^I(-)^
Plasmid-encoded regulator (Per)	-	-	perC/bfpW *^(-)^	perC/bfpW *^(-)^	-
Cytolethal distending toxin (CDT)	cdtA *^(56.03%)^, cdtB *^(68.87%)^, cdtC *^(40.56%)^	-	-	-	cdtA *^(56.03%)^, cdtB *^(68.87%)^, cdtC *^(40.56%)^
Adhesive fimbriae	csnA ^IH(-)^, cswA ^I(-)^	csnA^I(-)^, cswA^IH(-)^	csnA ^I(-)^, cswA ^IH(-)^	cfaB *^(-)^, cooA *^(-)^, csbA ^IH(-)^, csnA *^(-)^, cswA ^IH(-)^	csnA ^IH(-)^, cswA ^I(-)^
Cytotoxic necrotizing factor 1 (CNF1)	cnf1 *^(53.48%)^	-	-	-	cnf1 *^(53.48%)^
P fimbriae	-	-	papE *^(-)^, papG *^(-)^, papJ *^(-)^	-	-

^(%)^ Indicates the percent identity (%ID) that the protein subunit shares with its virulent counterpart.^(-)^ indicates that a %ID was not calculated because the virulence factor was not present or lacked required subunits. ^I^ Gene was detected in the assembly from Illumina MiSeq. ^H^ Gene was detected in the hybrid assembly. * Gene was detected in every dataset.

## Data Availability

The completed genomes described within the text are available in a publicly accessible repository. NCBI GenBank accession numbers are located in Table 3.

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
