# Peer review of "Complete Whole Genome Sequences of Escherichia coli Surrogate Strains and Comparison of Sequence Methods with Application to the Food Industry"

_microorganisms, 2021, doi:10.3390/microorganisms9030608_

Round 1
Reviewer 1 Report
Citation should appear in the methodology when describing the BLAST method.
Add a "Conclusion" section. As this is an extensive article, it should be summed up. You can also write about the application of this study.
Besides, the article is very well written and extensive. Congratulations.
Author Response
Thank you for your helpful comments. The BLAST citation has been inserted, along with corrected reference to specific aspects of this tool – appreciate your catch of this oversight. The final conclusions section was also added.
Thanks again for the kind remarks.
Reviewer 2 Report
Therrien et al. compared the MinION long-read, Illumina short-read, and combined sequence assemblies for 5 E. coli strains. Although the manuscript is well organized and the results are present clearly, the comparison of different sequencing platforms had been addressed in many papers. I have only few minor comments as below,
- line 146, "the Rapid Barding Kit" should be "the Rapid Barcoding Kit".
- line 212, by only "100-200 kb" between the MinION and MiSeq sequencing platforms. should be 100-300 kb.
- line 370, "Table 6" is a incorrect label.
Author Response
Thank you for your careful review. We realize that the long-read, short-read, and hybrid assembly is not novel, but among those in the food safety community not well-versed in genetics, genomics, or sequence analysis, we have noted some confusion and misunderstanding, particularly around what “WGS” entails. We wanted to generate high quality reference genomes of these surrogates as a resource and to screen for virulence factor gene sequences. Because a few rif-resistant derivatives were available, we included analysis of the rifR genes in the annotation. We thought it would be of value to some of the community to carry out a full comparison of the results of the three types of analysis in some depth – in way that is not necessarily a how-to guide, but a manuscript that contributes useful new data that can serve as a reference and help generate insight.
We have corrected the 3 specific errors you identified.
Reviewer 3 Report
Therrien et al. describe in their manuscript the applicability of WGS for microbial identification and safety assessment in the food industry. As sequencing costs decline, this method might become an attractive tool for routine analysis. The study was performed on five E. coli strains that are currently used as surrogates in the food industry. The study is sound and interesting.
Major comments:
The aims of the study should be stated more clearly in abstract and introduction: is it the comparison of different sequencing techniques, the comparison of wild type strains with their rifR counterparts, or the sequence analysis with subsequent safety evaluation of the five surrogate strains? The possible application in the food industry – as stated in the title – is only mentioned very briefly at the end of the discussion.
In the reviewer’s opinion, WGS followed by bioinformatic analysis of microorganisms is the future way to analyze food-related microorganisms, e.g. surrogate microorganisms. However, one should keep in mind that negative results (virulence genes are not present) might be considered as safe, but positive results (virulence genes are present), even when the genes might look as incomplete or non -functional), should be verified by laboratory experiments as mentioned below.
The presence of plasmids in strains BAA-1428 and BAA-1430 was inferred from the fact that the number of contigs could not be reduced to one and that the remaining sequences matched to those of plasmids when a database search was performed. The presence of plasmids should at least be confirmed by a laboratory experiment: plasmid preparation and analysis by agarose gel electrophoresis.
- coli is known to possess regions of repetitive sequences which might pose challenges in sequence assembly. Was this problem also faced by the authors? Please include in the manuscript.
- coli Nissle was chosen as a reference for the determination of the virulence attributes, and rather low values of sequence identity were thus obtained, as one might expect. The identified genes should also be compared to those of respective pathogenic strains.
In their discussion, the authors speculate about the virulence of the surrogate strains and conclude that despite the presence of single virulence factors the strains do not pose a hazard. Selected results, such as the expression of toxins, should be confirmed by lab experiments. For example, the presence of adherence factor genes and CNF1, even when supposed it’s not functional, should be analyzed by a cell culture adherence and cytotoxicity experiment.
Minor comments:
L122-123: Please state more clearly why these three rifR strains were chosen and why not rifR mutants of all five strains were included in the study.
L269: Please refer already here to Table S3.
Figures S1 and S2: Please label X-axes.
Tables S1 and S2: Please combine to one table to facilitate an easy overview.
Author Response
Reviewer,
Thank you for a very thorough and thoughtful review. The careful and constructive comments have resulted in an improved manuscript.
R3: “Therrien et al. describe in their manuscript the applicability of WGS for microbial identification and safety assessment in the food industry. As sequencing costs decline, this method might become an attractive tool for routine analysis. The study was performed on five E. coli strains that are currently used as surrogates in the food industry. The study is sound and interesting.”
Thank you. We agree that WGS is the future and have modified the conclusions section to reflect some of your well-stated comments
R3: Major comments:
The aims of the study should be stated more clearly in abstract and introduction: is it the comparison of different sequencing techniques, the comparison of wild type strains with their rifR counterparts, or the sequence analysis with subsequent safety evaluation of the five surrogate strains? The possible application in the food industry – as stated in the title – is only mentioned very briefly at the end of the discussion.
Your points are well taken here. Among those in the food safety community not well-versed in genetics, genomics, or sequence analysis, we have noted some confusion and misunderstanding at times, particularly around what “WGS” is or entails. We wanted to generate high quality reference genomes of these surrogates as a resource and to screen for virulence factor gene sequences. Because rif-resistant derivatives from 3 of the surrogates were available, we decided that we should go ahead and include analysis of the rifR genes in the annotation. In addition to making the annotations available, we thought it would be of value to some of the community to carry out a full comparison of the results of the three types of analysis in some depth – in a way that is not necessarily a how-to guide, but a manuscript that contributes useful new data that can serve as a reference and help generate insight – particularly for people who utilize these organisms.
R3: In the reviewer’s opinion, WGS followed by bioinformatic analysis of microorganisms is the future way to analyze food-related microorganisms, e.g. surrogate microorganisms. However, one should keep in mind that negative results (virulence genes are not present) might be considered as safe, but positive results (virulence genes are present), even when the genes might look as incomplete or non -functional), should be verified by laboratory experiments as mentioned below.
Agreed. You make a very important point. We edited the description of the surrogate organisms in the methods to clarify that each strain had been screened for virulence prior to our analysis. A goal of this project was to contribute a genome sequence screen for known virulence factor subunit genes. In the case of those we identified, so many subunits were absent that, coupled with prior data, we do believe these are non-functional. However, we agree that these assumptions should be validated. We hope that the sequence data and accompanying annotations will contribute to understanding of genetic variation that alters virulence status.
R3: The presence of plasmids in strains BAA-1428 and BAA-1430 was inferred from the fact that the number of contigs could not be reduced to one and that the remaining sequences matched to those of plasmids when a database search was performed. The presence of plasmids should at least be confirmed by a laboratory experiment: plasmid preparation and analysis by agarose gel electrophoresis.
You make another good point here. Our analysis of the genome assemblies gave consistent, single contig genomes, except for the residual contigs. Because these could not be assembled into the genomes, even with high coverage, and hybrid assembly – lack of any common overlaps, etc., and because we identified nearly perfect matches with published plasmid sequences, we felt confident including these as putative plasmid sequences, but did not choose to pursue additional experiments at that time. Your recommendation would add confirmatory evidence. Unfortunately we were not able to carry out this experimental work in the laboratory during the past week. It would be a useful addition, but we do not think it would not change the sequencing findings. Given additional time to respond, we would be willing to conduct the plasmid preps if deemed necessary.
R3: coli is known to possess regions of repetitive sequences which might pose challenges in sequence assembly. Was this problem also faced by the authors? Please include in the manuscript.
Yes. The advantage of the Oxford Nanopore long read sequencing is that it can get through these problematic areas, and coupled with the short-read sequence for error correction the hybrid assembly manages to overcome difficult regions. Our analysis (line 244) indicates at least 99.8-99.9% completion of the genome, which we consider a high quality result. We added comments to include reference to repeat elements in lines 62 and 471 to make this more clear.. In addition, the prokaryotic annotation tool of GenBank inserts annotation of “features” in the genome sequence so the location of repeat elements is also annotated for these deposited sequences.
R3: coli Nissle was chosen as a reference for the determination of the virulence attributes, and rather low values of sequence identity were thus obtained, as one might expect. The identified genes should also be compared to those of respective pathogenic strains.
Oh, yes. coli Nissle was used as a filter. Because it is used as a probiotic, we consider it a useful choice for filtering out negative results – to eliminate “background noise.” This is done with the idea that by subtracting Nissle from our strains, we would not accidentally delete any potential virulence genes.
From there, as shown in Table 4, we queried our genome sequences for the full sequence of known virulence genes – characterized from pathogenic strains with functional virulence factors. So we did exactly as you ask and compared to pathogenic strain genes.
R3: In their discussion, the authors speculate about the virulence of the surrogate strains and conclude that despite the presence of single virulence factors the strains do not pose a hazard. Selected results, such as the expression of toxins, should be confirmed by lab experiments. For example, the presence of adherence factor genes and CNF1, even when supposed it’s not functional, should be analyzed by a cell culture adherence and cytotoxicity experiment.
Yes, we agree with you. The strains were confirmed to lack antibiotic resistance or a subset of known virulence factors by the E. coli Reference Center of Pennsylvania State University and underwent further toxin testing via the application of commercial kits as well as tissue culture testing (African green monkey kidney (Vero) cells) by the depositor (Line 113)
Minor comments:
R3: L122-123: Please state more clearly why these three rifR strains were chosen and why not rifR mutants of all five strains were included in the study.
Good point. These strains were not specifically chosen – they were available in the Taylor lab from another, completely unrelated project and donated to this project. We simply as an opportunity to include annotation of the rif resistance genes, but did not think it was within the scope of the project to derive additional mutants. We revised line 129 to make this more clear.
R3: L269: Please refer already here to Table S3.
Corrected and update to reflect revised tables.
R3: Figures S1 and S2: Please label X-axes.
Corrected.
R3: Tables S1 and S2: Please combine to one table to facilitate an easy overview.
For tables S1 & S2 the phrase “contigs” was added as the x-axes. We sought to show a visual demonstration of the size distribution of individual contig lengths between the surrogate counterparts based on which sequencer was used. For example in Figure S1a. the Y-axis represents the size of the contigs while the x axis represents the total number of contigs in this case contig 1-74. We did not include the individual numbers on the X axis because it would not be informative. However, we could include an excel document as supplemental data if desired that would list the exact size of each contig in combination if that would be beneficial. See the supplemental data for the correction.
Round 2
Reviewer 2 Report
I'm satisfied with the author's corrections.
Reviewer 3 Report
All relevant reviewers comments have been answered and I recommend publication.